# Impact of Subclinical Congestion on Outcome of Patients Undergoing Mitral Valve Surgery

**DOI:** 10.3390/biomedicines8090363

**Published:** 2020-09-19

**Authors:** Anne-Kristin Schaefer, Thomas Poschner, Martin Andreas, Alfred Kocher, Günther Laufer, Dominik Wiedemann, Markus Mach

**Affiliations:** Division of Cardiac Surgery, Medical University of Vienna, 1090 Vienna, Austria; anne-kristin.schaefer@meduniwien.ac.at (A.-K.S.); thomasposchner@hotmail.com (T.P.); martin.andreas@meduniwien.ac.at (M.A.); alfred.kocher@meduniwien.ac.at (A.K.); guenther.laufer@meduniwien.ac.at (G.L.); dominik.wiedemann@meduniwien.ac.at (D.W.)

**Keywords:** plasma volume, subclinical congestion, heart failure, cardiac decompensation

## Abstract

Since risk assessment prior to cardiac surgery is based on proven but partly unsatisfactory scores, the need for novel tools in preoperative risk assessment taking into account cardiac decompensation is obvious. Even subclinical chronic heart failure is accompanied by an increase in plasma volume. This increase is illustrated by means of a plasma volume score (PVS), calculated using weight, gender and hematocrit. A retrospective analysis of 187 consecutive patients with impaired left ventricular function undergoing mitral valve surgery at a single centre between 2013 and 2016 was conducted. Relative preoperative PVS was generated by subtracting the ideal from actual calculated plasma volume. The study population was divided into two cohorts using a relative PVS score > 3.1 as cut-off. Patients with PVS > 3.1 had a significantly higher need for reoperation for bleeding/tamponade (5.5% vs. 16.7%; *p* = 0.016) and other non-cardiac causes (9.4% vs. 21.7%; *p* = 0.022). In-hospital as well as 6-month, 1-year and 5-year mortality was significantly increased in PVS > 3.1 (6.3% vs. 18.3%; *p* = 0.013; 9.4% vs. 23.3%; *p* = 0.011; 11.5% vs. 23.3%; *p* = 0.026; 18.1% vs. 33.3%; *p* = 0.018). Elevated PVS above the defined cut-off used to quantify subclinical congestion was linked to significantly worse outcome after mitral valve surgery and therefore could be a useful addition to current preoperative risk stratification.

## 1. Introduction

While contemporary cardiothoracic surgical risk models are mainly based on the evaluation of left ventricular ejection fraction, which is often overestimated in patients with higher degrees of mitral valve insufficiency, newly developed assessment tools for chronic heart failure and cardiac decompensation might substantially aid preprocedural risk assessment as well as planning and timing of the procedure.

As mitral regurgitation (MR) inherently leads to left ventricular dilatation via increased filling pressures and refractory volume overload, chronic heart failure (CHF) is a common concomitant factor of patients referred for surgical mitral valve repair with significant impact on postprocedural outcome [1,2,3,4]. A well-known consequence of CHF is an increase in plasma volume triggering acute decompensation with a significant impact on the prognosis of these patients [5,6]. Recent literature has demonstrated that plasma volume status (PVS) can be calculated easily based on the patients’ weight, gender and hematocrit [7]. Furthermore, an elevated PVS is known to directly correlate with increased mortality in patients with stable CHF, and has been applied for risk stratification in patients undergoing coronary artery bypass graft surgery or transcatheter aortic valve replacement [8,9].

This study aimed to assess (1) the impact of preoperative, potentially not clinically apparent heart failure on mortality and adverse events in patients undergoing surgical mitral valve repair/replacement and (2) the predictive value of PVS in preoperative risk assessment.

With survival rates often worse than patients harboring malignant diseases, patients with CHF are at great risk of adverse procedure related adverse events [5]. This is not only due to the underlying disease but also based on timing of the surgery. Compelling evidence has accumulated demonstrated that surgical treatment of mitral regurgitation is associated with considerably high post-operative and late mortality once symptoms have developed [10].

## 2. Materials and Methods

The retrospective analysis included a total of 187 patients with impaired left ventricular ejection fraction (LVEF ≤ 50%) undergoing mitral valve surgery with or without concomitant procedure at the Department of Cardiac Surgery, Medical University of Vienna between January 2013 and December 2016. The study was approved by the ethics committee of the Medical University of Vienna (EK1867/2019, approved on 3 September 2019), and informed consent was waived due to the retrospective study design.

Concomitant procedures are listed in Table 2 and included: coronary artery bypass grafting, tricuspid annuloplasty, atrial fibrillation surgery, left ventricle aneurysm surgery, left atrial appendage resection and patent foramen ovale closure.

LVEF was retrieved from routine preoperative echocardiogram reports. Baseline assessment prior to the surgical procedure included measurement of body weight and blood serum creatinine for calculation of the plasma volume status. The primary study endpoint was defined as 30-day mortality. Clinical outcome and the occurrence of related peri- and postprocedural complications were classified according to the updated Mitral Valve Academic Research Consortium (MVARC) criteria [11]. Long-term mortality data as well as the cause of death were retrieved by an inquiry to the Federal Institute of Statistics Austria.

### 2.1. Plasma Volume Equations

To calculate the actual plasma volume, an equation derived from curve fitting techniques using patients’ weight and hematocrit (Hct) compared to measurements from radioisotope assays was used [12].
actual PV = (1 – Hct) × [a + (b × weight (kg))])

The equation includes two constants to account for gender differences: a = 1530 for men and 864 for women; b = 41 for men and 47.9 for women.

The ideal plasma volume was calculated based on the following equation described by Longo et al. [13]:ideal PV = c × weight (kg)
with c being a constant that considers gender differences, corresponding to 39 for males and 40 for females. Subsequently, the patients’ percentual deviation from their ideal plasma volume was calculated [7].
PVS (%) = (actual PV − ideal PV)/ideal PV × 100

### 2.2. Statistical Analysis

Receiver operating characteristics (ROC) were used to assess the predictive accuracy of the PVS for the primary endpoint; cut-off values were calculated using the Youden Index. The study population was divided in two cohorts based on the relative PVS score (PVS ≤ 3.1 and PVS > 3.1). Based on their distribution, continuous variables were either expressed as median and interquartile range (IQR) or as mean and standard deviation (±SD) and compared using the Student’s *t*-test or the Mann-Whitney-U-test, respectively. Categorical variables were expressed as absolute numbers and percentage and compared with a Chi^2^-test or the Fisher’s exact test. Survival was visualized using Kaplan Meier curves and groups compared by the log-rank test.

A Cox proportional hazards model was used to examine the association between the PVS and the overall long-term mortality after mitral valve surgery and calculated hazard ratios and 95% confidence intervals. The period between surgery and either the last available follow-up or death was used to calculate the individual person-time interval. The hazard ratio was adjusted for baseline characteristics including the logistic EuroSCORE as well as the EuroSCORE II in a stepwise backward selection of factors based on their likelihood ratio and stratified by the PVS score.

The alpha level was set at <0.05; all reported *p*-values are two-sided. The statistical analyses were performed using SPSS, version 25.0 (IBM Corp, Armonk, NY, USA).

## 3. Results

### 3.1. Baseline Characteristics

Preoperative PVS was calculated in a total of 187 patients. If not stated otherwise, the second value refers to the higher PVS cohort. Preoperative medical history as well as the patients’ risk profile are depicted in Table 1.

About one third of patients presented in a preoperative state of (subclinical) heart failure according to PVS (PVS > 3.1; *n* = 60; 32.1%). The percentage of female patients in this cohort was strikingly lower (52.8% vs. 1.7%; *p* < 0.001). Patients showed no differences in age (66 ± 16 vs. 69 ± 14; *p* = 0.161), but higher risk scores (logistic EuroSCORE: 9.1 ± 13.1 vs. 13.7 ± 13.9; *p* = 0.047 and EuroSCORE II: 6.8 ± 8.6 vs. 10.3 ± 13.8; *p* = 0.004) as well as lower BMI (26.8 ± 6.1 vs. 24.4 ± 4.3; *p* = 0.004) could be found in the PVS > 3.1 cohort.

Increased morbidity in this cohort is reflected in a higher rate of insulin-dependent diabetes (3.1% vs. 13.3%; *p* = 0.012), chronic renal impairment (15.0% vs. 33.3%; *p* = 0.004) or worse renal function (preoperative creatinine (mg/dL) 1.0 ± 0.3 vs. 1.2 ± 0.8; *p* = 0.016 and preoperative creatinine clearance (mL/min) 70.6 ± 35.3 vs. 58.0 ± 41.8; *p* = 0.013). However, there was no significant difference in preoperative dialysis requirement (1.6% vs. 6.7%; *p* = 0.085). Furthermore, patients with a higher PVS were more symptomatic (NYHA IV: 11.8% vs. 28.3%; *p* = 0.001) and suffered more often from functional mitral regurgitation (51.2% vs. 68.3%; *p* = 0.019) and consequently also from preprocedural myocardial infarction (35.4% vs. 50.0%; *p* = 0.042) as well as coronary heart disease (55.1% vs. 75.0%; *p* = 0.007). The prevalence of arterial hypertension and COPD ≥ GOLD II was comparable in both groups (80.3% vs. 86.7%; *p* = 0.258 and 22.0% vs. 25.0%; *p* = 0.389, respectively).

Preoperative echocardiography demonstrated that patients with a PVS > 3.1 had a significantly lower left ventricular ejection fraction (40.0 ± 8.8 vs. 36.0 ± 10.0; *p* = 0.022). No difference in systolic pulmonary artery pressure ((in mmHg), 60.0 ± 34.0 vs. 61.0 ± 30.0; *p* = 0.449) was observed between the two PVS cohorts.

### 3.2. Alkaline Phosphatase Metabolism

Preoperative baseline alkaline phosphatase (AP) levels as well as absolute perioperative alkaline phosphatase loss were significantly higher in the high PVS group (Baseline AP, U/L: 67 ± 31 vs. 74 ± 36; *p* = 0.012; AP loss, U/L: 27 ± 17 vs. 33 ± 29; *p* = 0.012). AP on the first postoperative day (38 ± 21 vs. 43 ± 18; *p* = 0.178) showed no significant differences between the two cohorts.

### 3.3. Peri-and Postoperative Characteristics

The peri-and postoperative characteristics are outlined in Table 2. There was a significantly higher need for intraoperative blood products (59.1% vs. 75.0%; *p* = 0.024) in the PVS > 3.1 cohort, with both red blood cell units (1.7 ± 4.8 vs. 2.5 ± 3.0; *p* = 0.001) and platelet units (0.41 ± 2.7 vs. 0.42 ± 0.8; *p* = 0.027) being administered more frequently. Patients with a higher PVS required postoperative extracorporeal membrane oxygenation (ECMO) significantly more often (7.1% vs. 18.3%; *p* = 0.018). Furthermore, patients with a higher PVS score had a longer stay at the intensive care unit (( in days) 4.0 ± 7.0 vs. 6.0 ± 11.0; *p* = 0.015) as well as a strong trend in the total length of stay ((in days) 13 ± 12 vs. 15 ± 28; *p* = 0.063). Besides, those patients showed a significantly higher incidence of prolonged postprocedural ventilation (>24 h) (21.3% vs. 36.7%; *p* = 0.021) and a trend towards more frequent reintubation (4.7% vs. 11.7%; *p* = 0.079). No relevant differences regarding procedure types were found between the two cohorts.

### 3.4. Adverse Events and Survival

Adverse events are listed in Table 3. Incidence of reoperation due to bleeding or tamponade or for other non-cardiac causes was significantly higher in the high PVS group (bleeding/tamponade: 5.5% vs. 16.7%; *p* = 0.016; other non-cardiac causes: 9.4% vs. 21.7%; *p* = 0.22).

In-hospital mortality (6.3% vs. 18.3%; *p* = 0.013) was significantly higher in PVS > 3.1. Survival illustrated by Kaplan Meier curves (Figure 1) showed a significantly worse outcome in the higher PVS cohort (*p* = 0.014).

Adjusting the Cox proportionate hazards model for the logistic EuroSCORE and the EuroSCORE II, a significantly lower long-term survival of patients with a higher PVS was demonstrated (adjusted hazard ratio: 1.8; 95% CI: 1.0 to 3.4; *p* = 0.050; Figure 1, Table 4)

## 4. Discussion

Whereas about a third of all patients with reduced left ventricular function undergoing isolated or combined surgical mitral valve repair or replacement were in a state of clinical or subclinical decompensation at the time of the procedure, the present analysis demonstrates (I) that an elevated PVS above 3.1 is an independent predictor of in-hospital mortality in a multivariable analysis (II) that patients with an elevated PVS were prone to re-operations and prolonged postprocedural ventilation and (III) that a high preprocedural PVS is an indicator for impaired long-term survival over 5 years. Therefore, the calculation of the plasma volume status adds pivotal information to the timing of the procedure and the preoperative risk assessment of patients with heart failure prone to congestion undergoing mitral valve surgery.

While patients with severe symptomatic mitral regurgitation frequently suffer from recurrent cardiac decompensation [1], accompanying left ventricular dysfunction with subsequent pulmonary and systemic congestion is even more aggravated by the present valvular dysfunction [14,15] Therefore, a compensated preoperative state should be aimed for in every patient scheduled for elective valvular surgery whenever possible. However, the diagnosis and quantification of incipient, not yet clinically apparent cardiac decompensation and estimation of arising procedural risk can be challenging, and may not be sufficiently represented by contemporary, frequently applied risk scores.

The plasma volume status (PVS) as a surrogate parameter of congestion is known to directly correlate with increased mortality in patients with stable CHF and can be calculated using only three variables that are readily available for every patient undergoing cardiac surgery (hematocrit, bodyweight and gender) [6,7,8].

A state of congestion-not necessarily clinically apparent-is common in heart failure patients, associated with poorer outcomes, and bears the risk of perioperative exacerbation ultimately leading to acute decompensation. In line with findings in the known literature, 32.1% of patients in our cohort presented in a preoperative state of (subclinical) decompensation represented by a PVS above the defined cut-off level similar to those presented in recent studies. [6,7,9,16].

As patients within the high PVS group were more symptomatic, had a higher surgical risk profile and lower left-ventricular ejection fraction, they also had more complicated postoperative courses with significantly higher rates of ECMO support, longer stay at the intensive care unit, higher incidence of prolonged postprocedural ventilation, and more frequent bleeding or tamponade requiring re-sternotomy. In-hospital mortality was significantly higher in the group with a PVS > 3.1, and overall survival illustrated by Kaplan Meier curves was significantly worse in the higher PVS cohort (*p* = 0.014). Consequently, PVS calculation not only aids in correctly identifying a population at high risk for adverse events after mitral valve surgery, but also constitutes a benchmark in the preoperative preparation of these patients.

Conflicting data has been presented about the known gender difference with regard to the plasma volume status [6,7,17]. As (particularly elderly) women are known to be more prone to impaired LV diastolic reserve, the same amount of volume could result in different pressure levels as the left ventricle in women is often unable to mitigate the intravascular pressure due to the intrinsic diastolic properties [18,19]. Although the percentage of female patients in the high PVS group was lower in our analysis (only 1.7% vs. 52.8% in the low PVS group), the findings are in line with the gender differences in PVS calculation of previous studies [6,7]. A potential explanation for this finding in this particular context could be the lower absolute regurgitation volumes, as well as the smaller left atrial and ventricular dimensions of women compared to men when diagnosed with mitral regurgitation and referred to surgery [20].

Hemodilution and chronic iron deficiency anemia are common in patients with congestive heart failure and associated with adverse outcomes [21,22]. Furthermore, these factors may also be the underlying cause of lower baseline hematocrit levels and increased requirements of red blood cell transfusions observed in the high PVS group in the present study.

Chronic heart failure and associated multimorbidity, cachexia, and frailty might serve as a possible explanation for the significantly lower BMI found in the PVS > 3.1 cohort, and an impaired baseline end organ function in this cohort is reflected by worse renal function parameters in the high PVS group. Even though two-third of the patients with a high PVS suffered from functional mitral regurgitation and therefore had a higher prevalence of preprocedural myocardial infarction and coronary artery disease, rates in concomitant CABG procedures did not differ between the cohorts potentially indicating more advanced coronary artery disease without the possibility of revascularization accompanied by a pronounced ischemic cardiomyopathy component. One of the main factors contributing to a higher PVS in patients with functional mitral regurgitation is the significantly lower ejection fraction in this cohort. Consequently, these patients could carry the greatest benefit of a PVS-based recompensation strategy before undergoing cardiac surgical procedures.

Furthermore, we did not observe increased rates in atrial fibrillation surgery (e.g., MAZE) despite a higher rates of atrial fibrillation in the high PVS group. This might indicate that surgeons abstained from performing concomitant MAZE in these patients as a result of more advanced atrial dilation resulting from possibly more severe or longer history of mitral regurgitation.

In previous studies, increased perioperative alkaline phosphatase consumption has been linked to increased mortality and higher incidence of perioperative adverse events in patients undergoing cardiac surgery on cardiopulmonary bypass [23,24,25]. In line with these findings preoperative baseline alkaline phosphatase levels as well as absolute perioperative alkaline phosphatase consumption were significantly higher in the high PVS group in the present study due to plasma volume expansion and consecutively lower serum enzyme activity.

In conclusion, our study demonstrates that calculated PVS is a predictor for impaired survival and higher incidence of postoperative adverse events in patients undergoing mitral valve procedures, and a useful adjunct to concurrent risk stratification models. In contrast to current risk prediction models used in cardiac surgery, the PVS calculation is both a simple and quick tool to assess surgical risk and a dynamic parameter that can be modified with medical therapy. Therefore, the plasma volume status is a pivotal parameter during preoperative patient care prior and aids to correctly identify the temporal sweet spot for surgical mitral valve procedures. Nevertheless, further research in this area is needed since the effect of alterations of pre- or postoperative management (e.g., adaptation of medical heart failure therapy or preoperative iron repletion) has to be evaluated prospectively in larger trials.

### Limitations

Limitations inherent to every retrospective analysis apply to this study. Preoperative brain-natriuretic peptide was unavailable for a large proportion of patients and thus not correlated with preoperative PVS. While the PVS calculation helped identifying a high-risk population for adverse outcomes, preoperative plasma volume changes due to adapted medical therapy need to be assessed and validated in further research.

## Figures and Tables

**Figure 1 biomedicines-08-00363-f001:**
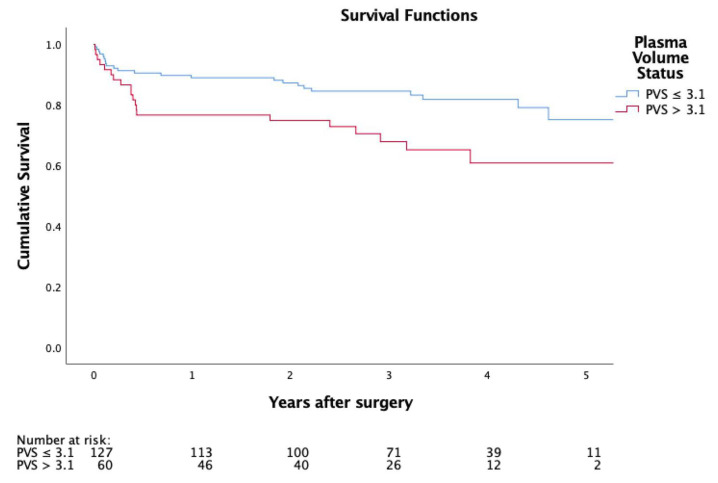
Kaplan-Meier curve representing postoperative survival.

**Table 1 biomedicines-08-00363-t001:** Patient Baseline Characteristics.

	Overall Cohort (*n* = 187)	PVS ≤ 3.1 (*n* = 127)	PVS > 3.1 (*n* = 60)	*p*-Value
Female, *n* (%)	68 (36.4)	67 (52.8)	1 (1.7)	0.000
Age, median (±IQR)	67.0 (15)	66.0 (16)	69.0 (14)	0.161
BMI, median (±IQR)	26.0 (5.5)	26.8 (6.1)	24.4 (4.3)	0.004
Logistic EuroSCORE, median (±IQR)	10.9 (13.2)	9.1 (13.1)	13.7 (13.9)	0.047
EuroSCORE II, median (±IQR)	7.6 (10.0)	6.8 (8.6)	10.3 (13.8)	0.004
Active smoker, *n* (%)	32 (17.1)	23 (18.1)	9 (15.0)	0.381
Chronic heart failure, *n* (%)	90 (48.1)	58 (45.7)	32 (53.3)	0.205
Hypertension, *n* (%)	154 (82.4)	102 (80.3)	52 (86.7)	0.258
Dyslipidemia, *n* (%)	106 (56.7)	68 (53.5)	38 (63.3)	0.482
Diabetes mellitus, *n* (%)	59 (31.6)	37 (29.1)	22 (36.7)	0.193
Diabetes mellitus (IDDM), *n* (%)	12 (6.4)	4 (3.1)	8 (13.3)	0.012
Chronic renal insufficiency, *n* (%)	39 (20.9)	19 (15.0)	20 (33.3)	0.004
Last preoperative creatinine (mg/dL), median (±IQR)	1.1 (0.4)	1.0 (0.3)	1.2 (0.8)	0.016
Preoperative Creatinine Clearance (mL/min), median (±IQR)	67.3 (38.1)	70.6 (35.3)	58.0 (41.8)	0.013
Preoperative dialysis, *n* (%)	6 (3.2)	2 (1.6)	4 (6.7)	0.085
Previous vascular stroke, *n* (%)	19 (10.2)	11 (8.7)	8 (13.3)	0.230
Neurological disease, *n* (%)	8 (4.3)	5 (3.9)	3 (5.0)	0.503
Prior myocardial infarction, *n* (%)	75 (40.1)	45 (35.4)	30 (50.0)	0.042
Coronary artery disease, *n* (%)	115 (61.5)	70 (55.1)	45 (75.0)	0.007
Prior CABG, *n* (%)	15 (8.0)	7 (5.5)	8 (13.3)	0.120
Prior PCI, *n* (%)	34 (18.2)	19 (15.0)	15 (25.0)	0.074
Prior valve surgery, *n* (%)	22 (11.8)	13 (10.2)	9 (15.0)	0.393
Thoracic aortic surgery *n* (%)	7 (3.7)	4 (3.1)	3 (5.0)	0.497
Atrial fibrillation, *n* (%)	95 (50.8)	66 (52.0)	29 (48.3)	0.379
AV-Block, *n* (%)	6 (3.2)	4 (3.1)	2 (3.3)	0.627
Prior pacemaker, *n* (%)	17 (9.1)	13 (10.2)	4 (6.7)	0.309
Prior ICD, *n* (%)	9 (4.8)	5 (3.9)	4 (6.7)	0.316
Endocarditis, *n* (%)	5 (2.7)	3 (2.4)	2 (3.3)	0.516
Liver cirrhosis, *n* (%)	1 (0.5)	0 (0.0)	1 (1.7)	0.321
NYHA class IV, *n* (%)	32 (17.1)	15 (11.8)	17 (28.3)	0.001
COPD Gold ≥ II, *n* (%)	43 (23.0)	28 (22.0)	15 (25.0)	0.389
Bronchodilators, *n* (%)	40 (21.4)	25 (19.7)	15 (25.0)	0.260
Left ventricular function, mean (±SD)	38.4 (9.3)	40.0 (8.8)	36.0 (10.0)	0.022
Severe mitral regurgitation, *n* (%)	161 (86.1)	110 (86.6)	51 (85.0)	0.850
Primary mitral regurgitation, *n* (%)	81 (43.3)	62 (28.8)	19 (31.7)	
Secondary mitral regurgitation, *n* (%)	106 (56.7)	65 (51.2)	41 (68.3)	
Moderate or severe tricuspid regurgitation, *n* (%)	65 (34.8)	44 (34.6)	21 (35.0)	0.883
Systolic pulmonary artery pressure in mmHg, median (±IQR)	60.0 (34)	60.0 (34.0)	61.0 (30.0)	0.449
Hematocrit, mean (±SD)	37.8 (5.2)	39.3 (5.1)	34.8 (4.2)	0.262
Preoperative alkaline phosphatase (AP) U/L, median (±IQR)	69.0 (34)	67.0 (31)	73.5 (36)	0.012
AP 1^st^ post-op day U/L, median (±IQR)	39.0 (20)	38.0 (21)	42.5 (18)	0.178
AP 1^st^ post-op day/preoperative AP %, median (±IQR)	59.6 (17.2)	60.3 (17.7)	56.3 (16.2)	0.065
Consumption of AP in U/L, median (±IQR)	27.0 (20)	27.0 (17)	33.0 (29)	0.012
Time between PVS calculation and surgery in d, median (±IQR)	2.0 (3)	2.0 (3)	3.0 (3)	0.265

Abbreviations: AP, Alkaline Phosphatase; CABG, Coronary Artery Bypass Graft; COPD, Chronic Obstructive Pulmonary Disease; eGFR, estimated Glomerular Filtration Rate; EuroSCORE, European System for Cardiac Operative Risk Evaluation; ICD–Implantable Cardioverter Defibrillator; IDDM, Insulin-Dependent Diabetes Mellitus; IQR, Interquartile Range; LVEF, Left Ventricular Ejection Fraction; NYHA, New York Heart Association; PCI, Percutaneous Coronary Intervention; sPAP, systolic Pulmonary Artery Pressure.

**Table 2 biomedicines-08-00363-t002:** Procedural Characteristics of the Cohort.

	Overall Cohort (*n* = 187)	PVS ≤ 3.1 (*n* = 127)	PVS > 3.1 (*n* = 60)	*p*-Value
Urgent operation, *n* (%)	56 (29.9)	32 (25.2)	24 (40.0)	0.089
Cardiogenic shock, *n* (%)	5 (2.7)	2 (1.6)	3 (5.0)	0.189
Isolated mitral valve repair, *n* (%)	17 (9.1)	14 (11.0)	3 (5.0)	0.520
Combined mitral valve repair and CABG, *n* (%)	47 (25.1)	28 (22.0)	19 (31.7)	0.520
Isolated mitral valve replacement, *n* (%)	7 (3.7)	5 (3.9)	2 (3.3)	0.520
Combined mitral valve replacement and CABG, *n* (%)	10 (5.3)	7 (5.5)	3 (5.0)	0.520
Combined mitral and atrial fibrillation surgery, *n* (%)	41 (21.9)	34 (26.8)	7 (11.7)	0.014
Minimal invasive mitral valve procedure, *n* (%)	8 (4.3)	6 (4.7)	2 (3.3)	0.497
LV aneurysm surgery, *n* (%)	3 (1.6)	2 (1.6)	1 (1.7)	0.678
Cardiopulmonary bypass in min, mean (±SD)	176.7 (60.4)	172.5 (57.9)	185.4 (65.2)	0.601
Aortic cross clamp time in min, mean (±SD)	107.4 (35.4)	107.4 (34.7)	107.6 (37.2)	0.847
Intraoperative blood products, *n* (%)	120 (64.1)	75 (59.1)	45 (75.0)	0.024
Intraoperative red blood cell units, mean (±SD)	2.0 (4.3)	1.7 (4.8)	2.5 (3.0)	0.001
Intraoperative fresh frozen plasma units, mean (±SD)	0.7 (2.2)	0.5 (1.6)	1.1 (3.0)	0.089
Intraoperative platelet units, mean (±SD)	0.41 (2.3)	0.41 (2.7)	0.42 (0.8)	0.027
Postoperative blood products, *n* (%)	55 (29.4)	35 (27.6)	20 (33.3)	0.261
Postoperative red blood cell units, mean (±SD)	1.0 (3.2)	0.8 (2.3)	1.4 (4.6)	0.552
Postoperative fresh frozen plasma units, mean (±SD)	0.2 (1.3)	0.2 (0.8)	0.4 (2.1)	0.710
Postoperative platelet units, mean (±SD)	0.09 (0.6)	0.04 (2.6)	0.18 (1.1)	0.336
Implanted intraaortic balloon pump, *n* (%)	1 (0.5)	0 (0.0)	1 (1.7)	0.321
Implanted ECMO, *n* (%)	20 (10.7)	9 (7.1)	11 (18.3)	0.018
Reintubation, *n* (%)	13 (7.0)	6 (4.7)	7 (11.7)	0.079
Length of stay at ICU (total), median (±IQR)	5.0 (8.0)	4.0 (7.0)	6.0 (11.0)	0.015
Readmission at ICU, *n* (%)	13 (7.0)	7 (5.5)	6 (10.0)	0.204

Abbreviations other than in Table 1: LV, Left Ventricle; ECMO, Extracorporal Membrane Oxygenation; ICU, Intensive Care Unit.

**Table 3 biomedicines-08-00363-t003:** Postoperative Adverse Events and Outcome.

	Overall Cohort (*n* = 187)	PVS ≤ 3.1 (*n* = 127)	PVS > 3.1 (*n* = 60)	*p*-Value
Neurological adverse events				
Transient ischemic attack, *n* (%)	1 (0.5)	0 (0.0)	1 (1.7)	0.321
Postoperative stroke ≥ 72 h, *n* (%)	7 (3.7)	4 (3.1)	3 (5.0)	0.400
Continuous Coma ≥ 24 h, *n* (%)	4 (2.1)	3 (2.4)	1 (1.7)	0.615
Other neurological complications, *n* (%)	15 (8.0)	10 (7.9)	5 (8.3)	0.560
Renal failure				
Acute Kidney Injury Stage III, *n* (%)	16 (8.6)	10 (7.9)	6 (10.0)	0.408
Postoperative hemofiltration, *n* (%)	13 (7.0)	8 (6.3)	5 (8.3)	0.408
Conduction disturbances				
New AV-Block III, *n* (%)	10 (5.3)	7 (5.5)	3 (5.0)	0.594
New Atrial Fibrillation, *n* (%)	36 (19.3)	23 (18.1)	13 (21.7)	0.349
Pulmonary adverse events				
Prolonged ventilation (>24 h), *n* (%)	49 (26.2)	27 (21.3)	22 (36.7)	0.021
Pneumonia, *n* (%)	20 (10.7)	11 (8.7)	9 (15.0)	0.146
Pulmonary embolism, *n* (%)	1 (0.5)	0 (0.0)	1 (1.7)	0.679
Miscellaneous adverse events				
Acute peripheral ischemia, *n* (%)	2 (1.1)	1 (0.8)	1 (1.7)	0.540
Complication of anticoagulation, *n* (%)	5 (2.7)	3 (2.4)	2 (3.3)	0.516
Gastrointestinal complication, *n* (%)	6 (3.2)	5 (3.9)	1 (1.7)	0.373
Perioperative myocardial infarction, *n* (%)	0 (0.0)	0 (0.0)	0 (0.0)	*n*.s.
Cardiac tamponade, *n* (%)	0 (0.0)	0 (0.0)	0 (0.0)	*n*.s.
Aortic dissection, *n* (%)	0 (0.0)	0 (0.0)	0 (0.0)	*n*.s.
Multiorgan failure, *n* (%)	9 (4.8)	4 (3.1)	5 (8.3)	0.121
Cardiac arrest, *n* (%)	15 (8.0)	9 (7.1)	6 (10.0)	0.337
Reoperations				
Due to bleeding/tamponade, *n* (%)	17 (9.1)	7 (5.5)	10 (16.7)	0.016
Due to valve dysfunction, *n* (%)	5 (2.7)	3 (2.4)	2 (3.3)	0.516
Due to other cardiac reason, *n* (%)	36 (19.3)	22 (17.3)	14 (23.3)	0.218
Due to other non-cardiac reason, *n* (%)	25 (13.4)	12 (9.4)	13 (21.7)	0.022
Length of stay in days, median (±IQR)	13.0 (13)	13.0 (12)	15.0 (28)	0.063
Hospital mortality *n* (%)	19 (10.2)	8 (6.3)	11 (18.3)	0.013
30-day all-cause mortality, *n* (%)	8 (4.3)	4 (3.1)	4 (6.7)	0.229
Hospital readmission within 30 days, *n* (%)	10 (5.3)	4 (3.1)	6 (10.0)	0.059

Abbreviations other than in Table 1 and Table 2: h–hours. *n*.s.= non-significant

**Table 4 biomedicines-08-00363-t004:** Multivariate Cox Proportionate Hazards Model of Predictive Factors for Long-Term Mortality after Mitral Valve Surgery

	Multivariate Analysis		
	OR	95% CI	*p*-Value
Demographics			
Age	1.002	0.987–1.017	0.766
Gender	1.313	0.926–1.861	0.126
Preoperative alkaline phosphatase	1.001	0.995–1.008	0.658
Logistic EuroSCORE	1.010	0.992–1.028	0.268
EuroSCORE II	0.980	0.950–1.012	0.218
Procedure type	1.44	0.969–2.150	0.071
PVS > 3.1	1.833	0.999–3.361	0.050

Abbreviations other as in (Table 1, Table 2 and Table 3): CI–Confidence Interval; OR, Odds Ratio.

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
