# Peer review of "Impact of Subclinical Congestion on Outcome of Patients Undergoing Mitral Valve Surgery"

_biomedicines, 2020, doi:10.3390/biomedicines8090363_

Round 1
Reviewer 1 Report
This work is of interest since it offers a simple formula to estimate PVS.
There are some methodological questions.
Functional and structural mitral valve regurgitation should be separated. It is unclear if this has been done.
A reference for the formula (section 4.1) must be added.
As the authors state, the iron status plays an important role in CHF. Are these results available?
Is PVS the only predictor for outcome? If not, a table should show all predictors, hence their clinical importance can be compared.
Author Response
Response to reviewers
First and foremost, we would like to thank you for your insightful comments, your fruitful suggestions and the opportunity to revise our manuscript. All comments were revised according to the reviewer`s comments and are listed and answered below point-by-point; changes are highlighted in the text.
Reviewer 1:
This work is of interest since it offers a simple formula to estimate PVS.
There are some methodological questions.
- Functional and structural mitral valve regurgitation should be separated. It is unclear if this has been done.
Answer: Thank you for the valuable remark. This issue was addressed in the revision. The stratification in FMR and non-FMR patients revealed that patients with a higher PVS suffered more frequently from secondary MR. The most plausible and obvious explanation for this finding is the lower ejection fraction in this cohort. However, this might also constitute a patient population that could carry the greatest benefit of a PVS-based recompensation strategy before undergoing cardiac surgery.
- A reference for the formula (section 4.1) must be added.
Answer: The reference has been added as requested.
- As the authors state, the iron status plays an important role in CHF. Are these results available?
Answer: Unfortunately, ferritin and transferrin levels are not included in preoperative routine baseline blood sample measurements. Due to the retrospective nature of the analysis, these data are not available.
- Is PVS the only predictor for outcome? If not, a table should show all predictors, hence their clinical importance can be compared.
Answer: The PVS was the only predictor for impaired long-term survival in our Cox-regression model. We decided to not include a specific table as we felt it wouldn’t add any additional information to the reader. Please let us kindly know if you want us to do so and we will follow your request without hesitation.
Reviewer 2 Report
The manuscript “Impact of subclinical congestion on outcome of patients undergoing mitral valve surgery” intends to assess the impact of preoperative clinically not apparent chronic heart failure on short- and long-term mortality and adverse events in patients undergoing mitral valve surgery and to show the predictive value of plasma volume status (PVS) in assessment of preoperative risk. The authors revealed that patients with PVS above 3.1 had higher in-hospital and long-term mortality and need for reoperation than patients with values of PVS bellow 3.1. The authors concluded that elevated PVS indicating the subclinical hypervolemia and congestion was associated with worth outcome after mitral valve surgery and may serve as a useful tool precising preoperative risk stratification.
Comments:
The topic is attractive, the work presents interesting data, and the text is generally well written. It seems to be fair that the authors admitted the limitations of the work in the limitation section of the discussion.
There are several minor comments which should be addressed:
It is not clear in the abstract, what does the a, b and c mean in the formula used for calculation of PVS (it is explained only in method section).
Why the value 3.1 was taken as a cut of value?
It is understandable that higher PVS is related to increased short term (hospital) mortality. On the other hand, long term (1-year or 5-year mortality) may be also related to inappropriate postoperative treatment. The authors should discuss the matter.
Page 7, line 26- “chronic heart failure and associated multimorbidity, cachexia… might serve as a possible explanation for significantly lower BMI” - …cachexia explains lower BMI?
Page 7, line 33-35 – the message of this paragraph is not clear, it should be reformulated.
Reference 21 – it does not seem to be correctly written
Author Response
Response to reviewers
First and foremost, we would like to thank you for your insightful comments, your fruitful suggestions and the opportunity to revise our manuscript. All comments were revised according to the reviewers' comments and are listed and answered below point-by-point; changes are highlighted in the text.
Reviewer 2:
The topic is attractive, the work presents interesting data, and the text is generally well written. It seems to be fair that the authors admitted the limitations of the work in the limitation section of the discussion.
There are several minor comments which should be addressed:
- It is not clear in the abstract, what does the a, b and c mean in the formula used for calculation of PVS (it is explained only in method section).
Answer: Thank you for this comment. We absolutely agree, that this might be confusing. We decided to delete the formula altogether in the abstract for several reasons: 1) to keep the abstract concise 2) to improve the readability of the abstract and 3) to increase the readers interest of the reader for the topic and the manuscript.
Please let us kindly know if this would be also in your interest, or if you want us to include all the specific formulas and variables in the abstract in addition to the methods section.
- Why the value 3.1 was taken as a cut of value?
Answer: As stated in the section “statistical analysis” on page 19 lines 468-470, ROC curves were calculated to predict the predictive power of the PVS for the primary endpoint. Based on these ROC-curves cut-off levels were calculated using the Youden-Index. Based on this calculation, a cut-off of 3.1 demonstrated the highest sensitivity and specificity.
- It is understandable that higher PVS is related to increased short term (hospital) mortality. On the other hand, long term (1-year or 5-year mortality) may be also related to inappropriate postoperative treatment. The authors should discuss the matter.
Answer: In our retrospective analysis, patients with a PVS above the defined cut-off had a worse short and long-term outcome. Due to the inherent nature of any retrospective analysis the presence of unidentified confounding variables cannot be excluded. Even though It is possible that other factors than PVS have influenced the observed outcome, all patients were treated at the same institution and received the same standard of postoperative medical care. Thus, we believe it is unlikely that have a systematic bias resulting from differences in postoperative management. The higher rate of functional MR, coronary artery disease, overall surgical risk and lower ejection fraction might be the most prominent contributors to the impaired long-term survival of this cohort.
- Page 7, line 26- “chronic heart failure and associated multimorbidity, cachexia… might serve as a possible explanation for significantly lower BMI” - …cachexia explains lower BMI?
Answer: Cardiac cachexia is a common finding in heart failure patients. Therefore, we are not surprised that see that patients with an elevated PVS (and therefore advanced heart failure state) demonstrate overall lower BMI levels. Please let us know if you want us to add a more specific explanation for this finding in the manuscript.
- Page 7, line 33-35 – the message of this paragraph is not clear, it should be reformulated.
Answer: Thank you for this remark. We reformulated this section in order to make it more understandable for the reader.
- Reference 21 – it does not seem to be correctly written
Answer: The reference was adapted accordingly.
Round 2
Reviewer 1 Report
A table with the predictors would add to the value of the manuscript since it shows the odds ratios (as a measure for clinical relevance) and p-values (as a measure for statistical significance). A non-significant value is also instructive
Author Response
A table with the predictors would add to the value of the manuscript since it shows the odds ratios (as a measure for clinical relevance) and p-values (as a measure for statistical significance). A non-significant value is also instructive
Answer: A table with the parameters included in the COX regression was added as requested. As the parameters were selected in a stepwise backward fashion based on their likelihood-ratio the respective OR as well as CI and p-values at the ultimate step before exclusion are displayed in table 4.
Please let us kindly know if this is to your satisfaction or if you require further details regarding our analysis.
Thank you for your valuable insights when reviewing our manuscript.